# Memories of *Memórias*: shaping a century of plague research and public health policy in Brazil

**Igor Vasconcelos Rocha/+, Matheus Filgueira Bezerra, Marise Sobreira, Alzira Maria Paiva de Almeida**

Fundação Oswaldo Cruz-Fiocruz, Instituto Aggeu Magalhães, Departamento de Microbiologia, Recife, PE, Brasil

Plague, caused by *Yersinia pestis*, remains a historically significant and reemerging zoonotic threat worldwide. Often erroneously considered a medieval relic, the disease persists in natural foci, including Brazil, where it was introduced in 1899 via maritime trade. Over the past 125 years, the country has experienced cyclical outbreaks concentrated in the northeast, where ecological conditions support enzootic transmission among wild rodents and their fleas. While improved surveillance and control have reduced human cases in recent decades, the pathogen's zoonotic nature and potential for rapid spread in a changing climate underscore its enduring public health relevance. *Memórias do Instituto Oswaldo Cruz* (MIOC), one of Latin America's foremost tropical medicine journals, has been instrumental in documenting and shaping the course of plague research in Brazil. Its archives provide a unique chronological record, from pioneering studies on serum production and reservoir ecology to modern molecular analyses of bacterial virulence. This perspective synthesises the seminal contributions published in *Memórias* that have defined our understanding of plague in Brazil, identifies critical knowledge gaps that persist, and discusses emerging challenges in an era of climate change and shifting zoonotic disease dynamics.

Key words: health surveillance - vector-borne diseases - *yersinia pestis* - plague - epidemiological studies

Plague, caused by the Gram-negative bacterium *Yersinia pestis*, represents one of history's most devastating infectious diseases, having caused three major pandemics that shaped human civilisation.[1] While often considered a relic of the past, plague persists as a reemerging zoonotic threat, with natural foci still active in Africa, Asia, and the Americas.[1,2] In Brazil, plague arrived through the port of Santos in 1899, eventually establishing endemic foci in the Northeast region's semi-arid areas, where it continues to circulate among rodent populations.[2,3] Despite significant progress in control measures, the disease's complex ecology, potential for rapid spread, and capacity to cause severe clinical manifestations (bubonic, septicaemic, and pneumonic forms) maintain its status as a public health concern.[1]

Over more than a century of continuous publication, *Memórias do Instituto Oswaldo Cruz* (MIOC) has served as a cornerstone for tropical disease research in Latin America,[4] including pivotal studies on *Y. pestis* and plague epidemiology in Brazil. The journal's archives chronicle the evolving scientific understanding of this pathogen, from pioneering serum therapy protocols[5] to contemporary molecular investigations of bacterial virulence and transmission dynamics.[2,6] These publications not only document Brazil's unique experience with plague but also contribute valuable insights to the global understanding of this ancient yet persistently relevant disease.

This comprehensive perspective has three principal objectives: first, to synthesise the key contributions of MIOC to our understanding of *Y. pestis* and plague dynamics in Brazil; second, to examine current challenges in surveillance, diagnosis, and control within Brazilian foci; and third, to identify emerging research priorities at the intersection of climate change, land use transformation, and advancing genomic technologies. By systematically analysing this corpus of plague research spanning 116 years, we demonstrate how the journal has not only documented but also driven scientific progress, serving as an unparalleled resource for understanding zoonotic disease management in tropical regions.

## The birth of Fiocruz and its pivotal role in Brazil's plague control

The dawn of the 20th century in Brazil was marked by profound public health challenges. The arrival of the bubonic plague in 1899, alongside outbreaks of yellow fever and smallpox, exposed the fragility of the nation's sanitary infrastructure and threatened its economic hubs, particularly its bustling ports.[7] It was within this crisis context that the Brazilian federal government initiated a campaign to modernise public health.[8] This effort culminated in the 1900 creation of the Instituto Soroterápico Federal[9] in the district of Manguinhos, Rio de Janeiro, under its first director, the Barão de Pedro Affonso. The institute would later be renamed Instituto

**doi:** 10.1590/0074-02760250269

**+ Corresponding author:** igor.rocha@fiocruz.br | ⓘ https://orcid.org/0000-0002-8599-6948

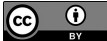

Oswaldo Cruz (IOC), in recognition of Oswaldo Cruz's role in Brazilian public health as the head of the Diretoria Geral de Saúde Pública, and become the cornerstone of the Fundação Oswaldo Cruz (Fiocruz).[10,11]

From its inception, the institute was thrust onto the front lines of the plague fight. Charged with producing immunobiologicals, the institute's first and most urgent mission was to manufacture a vaccine and serum against *Y. pestis*.[5,10,11] Overcoming immense technical difficulties and initial scepticism, Oswaldo Cruz and his team, which included key figures like Henrique Figueiredo de Vasconcellos, Antônio Cardoso Fontes, and Carlos Chagas,[12,13,14] successfully established large-scale production using equine hyperimmunisation. This achievement was nothing short of revolutionary for Brazil's public health autonomy, breaking the dependence on imported European products and ensuring a reliable supply of essential countermeasures for outbreak control.

The campaign against plague, however, extended far beyond the laboratory walls. Fiocruz scientists became instrumental in field epidemiology, investigating outbreaks and establishing the ecological foundations of the disease in the Brazilian context.[15]

The primary record of these endeavours is found in the pages of MIOC. Its early volumes are replete with original communications detailing the techniques for serum production, pathological findings from autopsies of human cases and experimental infections in animals, and meticulous descriptions of field surveys. These publications not only disseminated critical knowledge but also cemented the institute's scientific authority, showcasing a nascent national capability to confront a pandemic threat with rigour and innovation.

The legacy of this foundational period is profound. Fiocruz's direct involvement in plague control established a paradigm of integrating research, production, and public health intervention that defines the institution to this day.[2] The successful containment of urban plague outbreaks in major cities like Rio de Janeiro and Santos stands as a testament to its efficacy. By generating knowledge, producing life-saving tools, and deploying expertise into the field, Fiocruz emerged as the undisputed central pillar in Brazil's — and indeed, Latin America's — fight against plague, a role that its flagship journal, MIOC, has documented for over a century.[4]

## From serum therapy to molecular genetics: a century of plague studies in MIOC

The extensive corpus of plague research published in MIOC provides a unique and granular perspective on the evolution of scientific thought and practice in combating this disease in Brazil. This collection of 19 pioneering articles chronicles a clear trajectory from the foundational applied studies of the "Classical Bacteriology Era" to the integrative syntheses of the modern "Contemporary Era", all while maintaining a sharp focus on the specific epidemiological and ecological challenges of the Brazilian context (Figure). This documented evolution has produced a solid body of work instrumental in shaping the national approach to plague surveillance and control.

The inaugural study, published by MIOC in 1909, is a hallmark of the "Classical Bacteriology Era", where the priorities were isolation, characterisation, and the development of specific biological countermeasures. Developed by Vasconcellos, this work established a cornerstone for plague research in Brazil by detailing an experimental protocol for anti-plague serum production in horses.[5] The effort was quintessentially translational for its time, directly addressing the urgent public health need for locally produced biologicals and establishing Fiocruz's role as a hub for both discovery and production.[5]

However, the pages of MIOC also reveal that the institute's scientific function extended beyond reporting successes. Following the foundational 1909 article,[5] subsequent studies published in the journal reflected the intense and unresolved global scientific debate about the nature of the plague bacillus's pathogenicity. A study published in the journal exemplifies this by engaging directly with the period's central scientific controversy.[16] The research detailed a series of meticulous experiments that failed to isolate a free, filterable toxin from *Y. pestis* cultures using various chemical and physical methods. These results challenged the exotoxin theory of plague pathogenesis and underscored the

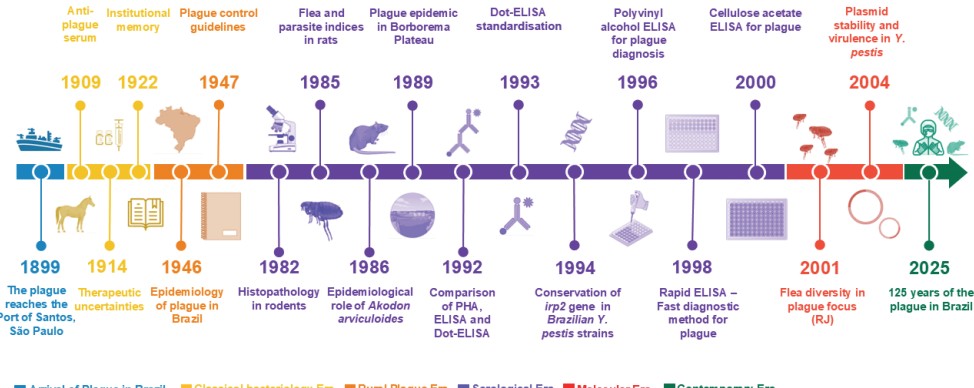

Timeline of plague research publications in *Memórias do Instituto Oswaldo Cruz*. The chart illustrates the evolution of scientific approaches to plague research in Brazil, categorised into five distinct eras based on the predominant methodological focus of the studies.

significant difficulties in standardising an immunising antigen,[16] thereby reflecting the profound unknowns that still surrounded the very mechanism of the disease and the therapy meant to combat it.

This foundational period of active research and debate was later curated and framed into a historical narrative by the journal itself. In 1922, the MIOC published a retrospective historical account by Vasconcellos.[17] By dedicating its pages to this work, the journal performed a crucial act of institutional memory, consciously shaping its own legacy. The article framed the early plague studies at Manguinhos, with their methodological innovations and collective self-experimentation, not merely as past events, but as the inaugural moment of a "true bacteriological technique" in Brazil.[17] This narrative thus served to canonise the institute's early trajectory, a period which, from its initial laboratory triumphs to its subsequent state-sponsored scientific expeditions, is identified by broader historical analyses as a pivotal chapter for the construction of modern Brazilian science.[18] Thus, the MIOC transformed the empirical struggles and debates it had previously documented into a coherent origin story, formally establishing the "Classical Bacteriology Era" within its own volumes as the foundational chapter in the history of Brazilian experimental science.

A significant leap in understanding the disease's patterns occurred with two seminal epidemiological studies from the mid-20th century, reflecting the transition into the "Rural Plague Era." This period was marked by a broadening of the public health arsenal against the disease, which came to involve integrated actions — from anti-rat campaigns and dichlorodiphenyltrichloroethane (DDT) pulverisations to the new possibilities of antimicrobial therapy. These tools began to shift the paradigm from purely immunological interventions towards a more multifaceted approach to control, particularly in non-urban settings. The first study, published by Barreto and de Castro,[19] systematically characterised the plague's profile in the Northeast through a comprehensive analysis of 2,610 cases. Its key findings established that the disease predominated in areas of precarious housing (slums), confirmed the bubonic form as the most prevalent, and identified *Xenopsylla cheopis* as the dominant flea vector in tropical zones. Crucially, it quantified a stark contrast in outcomes: while the overall lethality rate was 26%, early treatment with sulphonamides drastically reduced it to 12%, underscoring the vital importance of timely intervention.[19] Subsequently, a second study built upon this foundation to establish national control guidelines by historically evaluating measures adopted between 1936 and 1945.[20] It reinforced the predominance of the bubonic form while highlighting the extreme severity and high lethality of the pneumonic and septicaemic manifestations. Furthermore, it provided essential evidence for the efficacy of modern insecticides like DDT in controlling flea vectors and compounds like cyanogas for rodent extermination, offering a scientific basis for transitioning to these more effective prophylactic strategies and shaping the country's public health approach for decades. It is relevant to note that from the late 1920s through the 1970s, much of the broader Brazilian plague research was also published in parallel journals, particularly the *Archivos de Hygiene* and the *Revista Brasileira de Malariologia e Doenças Tropicais*, reflecting the expansive and collaborative effort required to combat the disease during this period.

From the 1980s onwards, a series of pivotal investigations marked the advent of the "Serological Era", characterised by a sophisticated use of immunology and refined field ecology to unravel the complex transmission cycles of plague.[21,22,23] Initial research was fundamental in identifying potential wild rodent reservoirs by characterising the histopathological markers of the infection, such as hepatic necrosis and splenic atrophy, in naturally infected animals.[21] Concurrently, complementary studies provided critical data for urban surveillance by assessing flea infestation indices, identifying seasonal peaks in *X. cheopis* prevalence that signalled periods of highest transmission risk.[22] A significant advancement came from detailed studies on specific sigmodontine rodents,[24] which revealed crucial ecological nuances by demonstrating that high flea infestation rates did not necessarily correlate with high susceptibility, indicating that not all abundant rodent species function as efficient reservoirs for the bacterium. The application of these integrated techniques was proven essential during the investigation of an active outbreak in the Borborema Plateau,[25] where bacteriological and serological diagnoses confirmed widespread *Y. pestis* circulation across 21 municipalities, with strains isolated from humans, domestic animals, and wild rodents. Further enriching this ecological understanding, a comprehensive survey in the Serra dos Órgãos Range provided the first complete inventory of the flea fauna in a distinct focus, documenting new host records and offering evidence of transmission bridges between wild and commensal rodent populations.[23]

A major and consistent thematic strand within this corpus has been the relentless refinement of diagnostic tools, marked by a focused effort to develop more sensitive, practical, and accessible serological assays. This endeavour began with a pivotal comparative study that evaluated three diagnostic tests, conclusively demonstrating the superior sensitivity of the Dot enzyme-linked immunosorbent assay (Dot-ELISA).[26] Building on this finding, subsequent research focused on standardisation and innovation. One study meticulously optimised key parameters of the Dot-ELISA to ensure specificity and reproducibility while eliminating false-positive reactions.[27] Simultaneously, researchers pioneered new solid phases to enhance practicality, first by creating a novel ELISA format using polyvinyl alcohol glutaraldehyde (PVA) discs, which proved highly effective for antibody detection in human samples.[28] A significant advancement in field applicability was then achieved with the development of a rapid PVA-ELISA, which dramatically reduced the total assay time from 36 h to just 3 h without substantial loss of sensitivity.[29] Further expanding the diagnostic toolkit, another study introduced cellulose acetate as a superior solid support. This material demonstrated enhanced performance over conventional plates and offered unique versatility by functioning effectively for both spectrophotometric

reading in a standard ELISA and visual interpretation in a dot-ELISA format.[30] Collectively, these studies epitomised the Serological Era. Through their publication, MIOC not only documented but actively disseminated this transition, moving beyond mere disease description to providing the Brazilian public health system with a robust, scalable, and practical diagnostic arsenal.

The dawn of the "Molecular Biology Era" in plague research was also captured within the journal's pages, introducing sophisticated genetic analysis to the Brazilian context. This began with a pivotal study that surveyed the virulence-associated *irp2* gene across strains isolated from multiple outbreaks.[31] Utilising DNA hybridisation, the research confirmed the gene's high conservation and highlighted a significant risk in strain maintenance as the gene could be lost after *in vitro* subculture.[31] Subsequently, a deeper investigation into genetic stability employed plasmid profiling and LD50 assays in mice to explore the relationship between plasmid content and virulence.[6] This first comparative study of its kind across multiple strains demonstrated that plasmid loss does not always directly correlate with a reduction in virulence, challenging a long-held assumption and revealing a more complex relationship between *Y. pestis* genetics and pathogenicity.

Finally, this extensive scientific legacy was masterfully synthesised in a comprehensive review that analysed the 125-year history of plague in Brazil,[2] representing the "Contemporary Era" of integrative science. This work meticulously documented the historical evolution of plague control, tracing the journey from the foundational efforts of the Oswaldo Cruz Era to the implementation of modern surveillance protocols. By analysing national plague service archives and historical documents, the review effectively contextualised the scientific contributions published in the journal within the broader arc of both national and global research. It underscored the enduring relevance of this rich historical data, arguing for its integration with contemporary molecular and digital technologies to build a more robust and sustainable surveillance system for the future.

The trajectory of publications in MIOC tells the story of Brazil's scientific battle against plague. It is a narrative of evolution: from serum therapy to molecular genetics, from descriptive outbreak reports to analytical epidemiology, and from basic diagnostic methods to rapid, field-friendly assays. Each article represents a building block, contributing to a comprehensive and sophisticated understanding of *Y. pestis* that remains vital for managing this ancient threat in the modern world.

## Current challenges and knowledge gaps

Despite this formidable trajectory of scientific progress documented in MIOC, plague persists as a public health challenge in Brazil, presenting a new set of complex obstacles that defy easy solutions.[32,33] The enzootic foci in the Northeastern, once the epicentre of human outbreaks, now presents a complex and silent threat. The primary challenge today is no longer managing large urban epidemics but understanding and monitoring the maintenance of *Y. pestis* in wild rodent populations to prevent its reemergence into human communities.[33]

A critical knowledge gap in Brazil, as in many endemic countries, is the precise understanding of the environmental and climatic drivers that govern the transition from enzootic stability to epizootic spread and subsequent spillover to humans.[33] Globally, studies have linked plague dynamics to fluctuations in temperature, rainfall, and humidity that affect rodent and flea populations.[34,35,36] In the Brazilian semi-arid region, characterised by extreme climatic variability and periodic droughts, the specific triggers for epizootics are not yet fully understood.[33,35] The historical lack of long-term, systematic ecological monitoring of reservoir species and their fleas in these foci has historically hindered the development of predictive models for outbreak risk. Although initiatives have begun to address this gap, the data series remain nascent.[33,36,37] Advancing our understanding of how environmental drivers — such as rainfall, vegetation, and altitude — interact with host and vector ecology is therefore critical for predicting the risk of new plague outbreaks in its natural foci. Furthermore, the ecological landscape of plague in Brazil is undergoing silent but profound transformations.[32,33] Changes in land use, agricultural expansion, and desertification are altering the habitats of known rodent reservoirs and may be facilitating the invasion of more adaptable species, potentially reshaping the transmission cycles.[32]

Another persistent challenge is the maintenance of diagnostic and surveillance capacity in the absence of human cases. The decline in reported incidents over recent decades has led to a "out of sight, out of mind" dilemma, risking a dangerous atrophy of clinical and laboratory expertise. Brazilian health professionals outside the historical foci may never encounter a plague case, leading to potential misdiagnosis and dangerous delays in treatment. This is a global issue, as seen in non-endemic countries where travel-related cases are often initially missed.

In Brazil, genomic studies have decisively answered critical questions by demonstrating that all circulating strains derive from a single introduction of the 1.ORI2 lineage during the Third Pandemic.[38,39] The current challenge, therefore, lies not in a lack of fundamental genomic knowledge, but in its application. The imperative is to implement sustained genomic surveillance within endemic foci. This ongoing effort is crucial to rapidly distinguish between local re-emergence and new introductions, predict outbreaks by integrating genomic data with ecological and climatic variables, and monitor for emerging antimicrobial resistance.[33,39]

## Perspectives and emerging research priorities

The rich legacy of plague research published in MIOC provides not only a historical record but also a robust foundation upon which to build the next century of inquiry. The future research agenda is a direct evolution of the thematic strands so meticulously chronicled in the journal's pages, pushing each into a new era of technological sophistication.

First, the paramount challenge remains decoding the silent maintenance of *Y. pestis* within enzootic foci. This pursuit is a direct extension of the foundational ecological studies published from the 1980s onwards, which first employed field trapping and histopathology to iden-

tify potential wild rodent reservoirs and characterise the histopathological markers of infection.[21,22,23,24,25] The next step is to leverage this field-based approach with modern tools: employing metagenomics on soil and flea samples to identify environmental reservoirs, and integrating long-term ecological data into powerful predictive models of spillover risk. This represents the natural evolution from descriptive ecology to predictive, quantitative science.

Second, the refinement of diagnostic tools must enter its molecular phase. The emerging priority is to build upon this legacy by developing rapid, field-deployable molecular diagnostics (*e.g.*, PCR, LAMP, and CRISPR-based platforms) that can detect the pathogen itself in remote areas, in addition to antibody detection to direct, genotype-based identification.

By sharing discoveries from these new frontiers, MIOC will continue to do what it has done for over a century: translate cutting-edge science into actionable knowledge. This ensures that the hard-won lessons of Brazil's past, so vividly documented in its pages, remain central to building a more resilient and proactive defence against this ancient yet ever-evolving threat.

## ACKNOWLEDGEMENTS

To the Brazilian National Plague Reference Service staff and to the Aggeu Magalhães Institute (FIOCRUZ-PE) for all furtherance. We also extend our thanks to the many colleagues and researchers whose dedicated work over the decades, much of it published in the pages of MIOC, built the rich legacy of plague research described here.

## AUTHORS' CONTRIBUTION

IVR and MFB - conceptualisation, investigation, data curation, and original draft preparation; MS - supervised the investigation, data curation, and reviewed the manuscript; AMPA - conceptualisation, supervision, validation, manuscript review and editing, and project administration. The authors declare no conflict of interest.

## DATA AVAILABILITY

The data supporting this study are available within the article.

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

# OPEN PEER REVIEW

Memórias do IOC thanks the anonymous reviewers for their contribution to the peer review of this work.

## FIRST REVIEW ROUND

REVIEWERS' COMMENTS

### REVIEWER #1

To the editors of Memórias do Instituto Oswaldo Cruz

It gave me great pleasure to read and review the article "Memories of Memórias: shaping a century of plague research and public health policy in Brazil".

Summary

The abstract is well written and matches the text; the article is original and relevant to the field of the history of plague in Brazil; its methodology is coherent; the references are adequate, although they could be expanded; and, regarding the figures, the timeline contains a small typo, as described below.

General Assessment

In this article, the authors review and discuss a set of 17 articles on plague published at the MIOC, from its first edition in 1909 until the present day. One of the most interesting insights of the authors is their periodisation, dividing their sources into five main eras: "Classical Bacteriology Era, Antibiotic Era, Serological Era, Molecular Era, and Contemporary Era." Drawing upon this periodisation, the authors then characterise each era, highlighting their main aspects and landmarks, the scientific shifts marking the end of the eras, and how the articles at the MIOC not only exemplify the eras' trends, but most importantly, how they structured each of these eras.

The authors are particularly suited to advance such a periodisation and analysis, as they wrote most of the articles of the last three periods of plague studies in Brazil. Therefore, I do not have any important comments on the Serological, Molecular, and Contemporary eras.

Conversely, I have a few comments regarding the historical background, the Classical Bacteriology Era, and the Antibiotic Era, as outlined below.

Historical Background

• There are a few historical mistakes that need to be fixed.

The authors wrote on page 6: "It was within this crisis context that the Brazilian federal government, under the leadership of President Rodrigues Alves and the visionary sanitarian Oswaldo Cruz, initiated a monumental campaign to modernize public health. This effort culminated in the 1900 creation of the Instituto Soroterápico Federal in the district of Manguinhos, Rio de Janeiro, which would later be renamed Instituto Oswaldo Cruz (IOC) and become the cornerstone of the Fundação Oswaldo Cruz (Fiocruz) (8,9)."

Comment: The reader can believe that Rodrigues Alves and Oswaldo Cruz were both behind the creation of the Instituto Soroterápico Federal, which is wrong. The President of Brazil in 1899-1900 was Campos Salle, and the first director of the Instituto Soroterápico Federal was the Barão de Pedro Affonso. Rodrigues Alves was elected in 1902 under the promise of "sanitising" Rio de Janeiro, to which he placed Oswaldo Cruz at the head of the Diretoria Geral de Saúde Pública. But this came after the creation of the Instituto Soroterápico Federal, and not before as the authors seem to suggest. Also, I would suggest removing the word "monumental" from the paragraph above. The modernisation of public health pushed by Rodrigues Alves and Oswaldo Cruz was important, but very limited to the city of Rio de Janeiro. Therefore, I suggest reworking this paragraph.

Again, on page 6: "From its inception, the institute was thrust onto the front lines of the plague fight. Charged with producing immunobiologicals, the institute's first and most urgent mission was to manufacture a vaccine and serum against Y. pestis (4,8,9). Overcoming immense technical difficulties and initial scepticism, Oswaldo Cruz and his team, which included key figures like Vital Brazil, Carlos Chagas, Emílio Ribas, and Adolfo Lutz (10), successfully established large-scale production using equine hyperimmunization."

Comment: The names cited in this phrase were not exactly part of Oswaldo Cruz's team in charge of producing the anti-plague serum or vaccine, and the real collaborators of Oswaldo Cruz – Henrique Figueiredo de Vasconcellos and Cardoso Fontes – were not mentioned in this sentence. Carlos Chagas is correct, as he joined Manguinhos in the 1900s, but Vital Brazil, Emilio Ribas and Adolpho Lutz were part of São Paulo Public Health bureaucracy, and although they were friends with Oswaldo Cruz, and even some of them latter joined Manguinhos, I do not think this would be fair to simply put them as part of the "team" of Oswaldo Cruz, especially in this first moment of serum production. Therefore, I suggest reworking this paragraph.

Classical Bacteriology Era

• In general, I found this era less developed than the others. Therefore, I would suggest expanding it a bit further by discussing two points: 1) MIOC was also a platform to discuss the tensions and unknowns of the anti-plague serum therapy, as evidenced by Arthur Moses's 1914 article. 2) In 1922, Figueiredo de Vasconcellos wrote one of the first histories of plague in Brazil at the MIOC, arguing that studies on plague started a new scientific moment in Brazil. Both articles should be mentioned.

Antibiotic Era/Antimicrobial Era.

• To start with, the authors use two terms to name this era: antimicrobial era in the text and antibiotic era on the timeline. I suggest choosing one and keeping it.

• But I do not think that either is the best term to describe this era of plague studies. Sure, antibiotics were a game-changer, but the articles at the MIOC were more interested in discussing how to fight plague, which involved several actions, from anti-rat campaigns to DDT pulverisations. Moreover, these articles were concerned on how to fight plague in rural areas. Therefore, I suggest something like the Rural Plague Era.

• Maybe the authors should mention, in a footnote, that the MIOC was not the main platform to publish about plague from the late 1920s until the 1970s. In this long period, two other journals were preferred by Brazilian and foreign scholars writing about plague in Brazil: the Archivos de Hygiene (in its different iterations) and the Revista Brasileira de Malariologia. To say that does not compromise the authors' argument, but it will help the readers to look for other journals if they want to expand their comprehension of plague history in Brazil.

Final Remarks

These small mistakes and omissions are minor and can be easily fixed. Once they are done, I recommend the publication of the article.

**REVIEWER #2**

The article presents all the aspects to publish in Memorias do Instituto Oswaldo Cruz.

For the referecences we suggest  increase some papers published in "História, ciências, Saúde" (https://www.scielo.br/j/hcsm/grid ).

We strongly recommend some papers about scientifical missions in colonial and republican Brazil and the articles of Nisia Trindade Lima. The following links can be used for research:

https://search.scielo.org/?q=*&lang=pt&count=15&from=0&output=site&sort=&format=sum-mary&fb=&page=1&filter%5Bjournal_title%5D%5B%5D=Hist%C3%B3ria%2C+Ci%C3%AAn-cias%2C+Sa%C3%BAde-Manguinhos&q=*Miss%C3%B5es&lang=pt&page=1

https://www.scielo.br/j/hcsm/a/ssftpHJTrFMGJRkvg83nrYm/?lang=pt

**AUTHORS' RESPONSE TO THE REVIEWERS**

Response to the reviewers

Reviewer: 1

It gave me great pleasure to read and review the article "Memories of Memórias: shaping a century of plague research and public health policy in Brazil".

The abstract is well written and matches the text; the article is original and relevant to the field of the history of plague in Brazil; its methodology is coherent; the references are adequate, although they could be expanded; and, regarding the figures, the timeline contains a small typo, as described below.

In this article, the authors review and discuss a set of 17 articles on plague published at the MIOC, from its first edition in 1909 until the present day. One of the most interesting insights of the authors is their periodisation, dividing their sources into five main eras: "Classical Bacteriology Era, Antibiotic Era, Serological Era, Molecular Era, and Contemporary Era." Drawing upon this periodisation, the authors then characterise each era, highlighting their main aspects and landmarks, the scientific shifts marking the end of the eras, and how the articles at the MIOC not only exemplify the eras' trends, but most importantly, how they structured each of these eras.

The authors are particularly suited to advance such a periodisation and analysis, as they wrote most of the articles of the last three periods of plague studies in Brazil. Therefore, I do not have any important comments on the Serological, Molecular, and Contemporary eras.

Conversely, I have a few comments regarding the historical background, the Classical Bacteriology Era, and the Antibiotic Era, as outlined below.

**Author:** Please accept our sincerest gratitude for your thorough review and comments on our manuscript. We greatly appreciate the time and expertise you dedicated to our work. We have carefully considered all your suggestions and have incorporated them into the revised text. Below, you will find our detailed point-by-point responses, outlining how we have addressed each of your specific comments. Once again, thank you for your essential contribution to improving our work.

Reviewer: 1
Historical Background
**1.** There are a few historical mistakes that need to be fixed.

The authors wrote on page 6: "It was within this crisis context that the Brazilian federal government, under the leadership of President Rodrigues Alves and the visionary sanitarian Oswaldo Cruz, initiated a monumental campaign to modernize public health. This effort culminated in the 1900 creation of the Instituto Soroterápico Federal in the district of Manguinhos, Rio de Janeiro, which would later be renamed Instituto Oswaldo Cruz (IOC) and become the cornerstone of the Fundação Oswaldo Cruz (Fiocruz) (8,9)."

The reader can believe that Rodrigues Alves and Oswaldo Cruz were both behind the creation of the Instituto Soroterápico Federal, which is wrong. The President of Brazil in 1899-1900 was Campos Salle, and the first director of the Instituto Soroterápico Federal was the Barão de Pedro Affonso. Rodrigues Alves was elected in 1902 under the promise of "sanitising" Rio de Janeiro, to which he placed Oswaldo Cruz at the head of the Diretoria Geral de Saúde Pública. But this came after the creation of the Instituto Soroterápico Federal, and not before as the authors seem to suggest. Also, I would suggest removing the word "monumental" from the paragraph above. The modernisation of public health pushed by Rodrigues Alves and Oswaldo Cruz was important, but very limited to the city of Rio de Janeiro. Therefore, I suggest reworking this paragraph.

**Author:** We sincerely thank the reviewer for this correction. The chronological point raised is crucial for historical accuracy, and we have thoroughly revised the paragraph to ensure the sequence of events is correctly presented. As suggested, we have also removed the word "monumental". The name of the Barão de Pedro Affonso has been appropriately included. The corrected text now reads: "It was within this crisis context that the Brazilian federal government initiated a campaign to modernize public health (8). This effort culminated in the 1900 creation of the Instituto Soroterápico Federal (9) in the district of Manguinhos, Rio de Janeiro, under its first director, the Barão de Pedro Affonso. The institute would later be renamed Instituto Oswaldo Cruz (IOC), in recognition of Oswaldo Cruz's role in Brazilian public health as the head of the Diretoria Geral de Saúde Pública, and become the cornerstone of the Fundação Oswaldo Cruz (Fiocruz) (10,11)".

**2.** Again, on page 6: "From its inception, the institute was thrust onto the front lines of the plague fight. Charged with producing immunobiologicals, the institute's first and most urgent mission was to manufacture a vaccine and serum against Y. pestis (4,8,9). Overcoming immense technical difficulties and initial scepticism, Oswaldo Cruz and his team, which included key figures like Vital Brazil, Carlos Chagas, Emílio Ribas, and Adolfo Lutz (10), successfully established large-scale production using equine hyperimmunization."

The names cited in this phrase were not exactly part of Oswaldo Cruz's team in charge of producing the anti-plague serum or vaccine, and the real collaborators of Oswaldo Cruz – Henrique Figueiredo de Vasconcellos and Cardoso Fontes – were not mentioned in this sentence. Carlos Chagas is correct, as he joined Manguinhos in the 1900s, but Vital Brazil, Emilio Ribas and Adolpho Lutz were part of São Paulo Public Health bureaucracy, and although they were friends with Oswaldo Cruz, and even some of them latter joined Manguinhos, I do not think this would be fair to simply put them as part of the "team" of Oswaldo Cruz, especially in this first moment of serum production. Therefore, I suggest reworking this paragraph.

**Author:** We thank the reviewer for this correction. We have reworked the paragraph as suggested. The sentence now correctly credits Oswaldo Cruz's immediate collaborators. The revised text reads: "Overcoming immense technical difficulties and initial scepticism, Oswaldo Cruz and his team, which included key figures like Henrique Figueiredo de Vasconcellos, Antônio Cardoso Fontes, and Carlos Chagas (12–14), successfully established large-scale production using equine hyperimmunization".

**3.** Classical Bacteriology Era. In general, I found this era less developed than the others. Therefore, I would suggest expanding it a bit further by discussing two points: **1)** MIOC was also a platform to discuss the tensions and unknowns of the anti-plague serum therapy, as evidenced by Arthur Moses's 1914 article. **2)** In 1922, Figueiredo de Vasconcellos wrote one of the first histories of plague in Brazil at the MIOC, arguing that studies on plague started a new scientific moment in Brazil. Both articles should be mentioned.

**Author:** We are grateful for this excellent suggestion. In direct response to point 1, we have expanded the discussion to show how the MIOC served as a platform for scientific debate. We incorporated an analysis of Arthur Moses's 1914 article and the uncertainties surrounding serum therapy at the time (please, see highlighted text in the main document). In direct response to point 2, we have significantly developed the narrative to include Figueiredo de Vasconcellos's 1922 historical account. We took the opportunity presented by this article to integrate a reference suggested by Reviewer 2, as its historical argument directly resonated with the broader historiography on scientific expeditions and nation-building. Specifically, we now show how Vasconcellos's article served to canonize the institute's early trajectory, thereby providing a primary source that anchors the later historical analysis of this formative period as a pivotal chapter for the construction of modern Brazilian science.

**4.** Antibiotic Era/Antimicrobial Era. To start with, the authors use two terms to name this era: antimicrobial era in the text and antibiotic era on the timeline. I suggest choosing one and keeping it. But I do not think that either

is the best term to describe this era of plague studies. Sure, antibiotics were a game-changer, but the articles at the MIOC were more interested in discussing how to fight plague, which involved several actions, from anti-rat campaigns to DDT pulverisations. Moreover, these articles were concerned on how to fight plague in rural areas. Therefore, I suggest something like the Rural Plague Era.

Maybe the authors should mention, in a footnote, that the MIOC was not the main platform to publish about plague from the late 1920s until the 1970s. In this long period, two other journals were preferred by Brazilian and foreign scholars writing about plague in Brazil: the Archivos de Hygiene (in its different iterations) and the Revista Brasileira de Malariologia. To say that does not compromise the authors' argument, but it will help the readers to look for other journals if they want to expand their comprehension of plague history in Brazil.

**Author:** We sincerely thank the reviewer for this insightful comment. We fully agree with the observation that the term antimicrobial/antibiotic Era does not fully capture the broader scope of the public health actions documented in the MIOC articles from this period. In direct response to this suggestion, we have adopted the proposed designation "Rural Plague Era" throughout the manuscript and in the timeline. Furthermore, we have incorporated the contextual note regarding other key publication platforms. We have included this information directly in the text to conclude the section's contextualization, rather than in a footnote. The added sentence reads: "It is relevant to note that from the late 1920s through the 1970s, much of the broader Brazilian plague research was also published in parallel journals, particularly the Archivos de Hygiene and the Revista Brasileira de Malariologia, reflecting the expansive and collaborative effort required to combat the disease during this period."

**5.** Final remarks. These small mistakes and omissions are minor and can be easily fixed. Once they are done, I recommend the publication of the article.

**Author:** We sincerely thank the reviewer for the positive final assessment and for the recommendation for publication.

Reviewer: 2

**1.** The article presents all the aspects to publish in Memorias do Instituto Oswaldo Cruz.

**Author:** We thank Reviewer 2 for the positive assessment. We have carefully considered the specific suggestions provided and have incorporated them into the revised manuscript.

**2.** For the referecences we suggest  increase some papers published in "História, ciências, Saúde" (https://www. scielo.br/j/hcsm/grid ) .

We strongly recommend some papers about scientifical missions in colonial and republican Brazil and the articles of Nisia Trindade Lima. The following links can be used for research: https://search.scielo. org/?q=*&lang=pt&count=15&from=0&output=site&sort=&format=summary&fb=&page=1&filter%5Bjournal_ti tle%5D%5B%5D=Hist%C3%B3ria%2C+Ci%C3%AAncias%2C+Sa%C3%BAde-Manguinhos&q=*Miss%C3%B5 es&lang=pt&page=1 and https://www.scielo.br/j/hcsm/a/ssftpHJTrFMGJRkvg83nrYm/?lang=pt.

**Author:** We thank the reviewer for the suggestion. We agree that these studies provide essential historical and historiographical context for understanding scientific missions and the construction of public health in Brazil. In direct response to the recommendation, we have incorporated three key references from this line of research into our bibliography:

**1.** We included the article "Não é meu intuito estabelecer polêmica": a chegada da peste ao Brasil, análise de uma controvérsia, 1899 (Nascimento and Silva, 2013), published in História, Ciências, Saúde – Manguinhos, that offers a detailed analysis of the public debate that preceded the creation of the Instituto Soroterápico Federal, contextualizing the political pressure for a scientific response.

**2.** We also added the study "A peste bubônica no Rio de Janeiro e as estratégias públicas no seu combate (1900-1906)", published in the journal Territórios e Fronteiras, by Dilene Raimundo do Nascimento, of the Casa de Oswaldo Cruz/Fiocruz, to better detail the public health crisis scenario following the arrival of the disease.

**3.** Specifically following the indication regarding the work of Nísia Trindade, we included her article "Missões civilizatórias da República e interpretação do Brasil", published in História, Ciências, Saúde – Manguinhos. We took the opportunity presented by the inclusion of Figueiredo de Vasconcellos's 1922 historical article (Reviewer 1) to draw a parallel between the internal narrative of the MIOC and broader historiographical interpretations of scientific expeditions as nation-building projects, citing this work.

We believe that incorporating these references in a contextual and integrated manner into our narrative addresses the spirit of the suggestion and adds a valuable layer of historical depth to our study, without diverting the primary focus of the analysis on the Memórias do Instituto Oswaldo Cruz periodical. We thank the reviewer again for the pertinent bibliographic guidance.

**SECOND REVIEW ROUND**

REVIEWERS' COMMENTS

**REVIEWER #1**

The authors have fully answered my comments, and I am pleased to recommend the acceptance of the article. Just a minor comment: notes 13 and 14 are the same, and one needs to be excluded. Otherwise, it looks fine.

**REVIEWER #2**

No comments.

