## [Reviewer Report · FIRST REVIEW ROUND - REVIEWERS COMMENTS]

## REVIEWER #1

To the editors of Memórias do Instituto Oswaldo Cruz

It gave me great pleasure to read and review the article “Memories of Memórias: shaping a century of plague research and public health policy in Brazil”.

Summary

The abstract is well written and matches the text; the article is original and relevant to the field of the history of plague in Brazil; its methodology is coherent; the references are adequate, although they could be expanded; and, regarding the figures, the timeline contains a small typo, as described below.

General Assessment

In this article, the authors review and discuss a set of 17 articles on plague published at the MIOC, from its first edition in 1909 until the present day. One of the most interesting insights of the authors is their periodisation, dividing their sources into five main eras: “Classical Bacteriology Era, Antibiotic Era, Serological Era, Molecular Era, and Contemporary Era.” Drawing upon this periodisation, the authors then characterise each era, highlighting their main aspects and landmarks, the scientific shifts marking the end of the eras, and how the articles at the MIOC not only exemplify the eras’ trends, but most importantly, how they structured each of these eras. The authors are particularly suited to advance such a periodisation and analysis, as they wrote most of the articles of the last three periods of plague studies in Brazil. Therefore, I do not have any important comments on the Serological, Molecular, and Contemporary eras. Conversely, I have a few comments regarding the historical background, the Classical Bacteriology Era, and the Antibiotic Era, as outlined below.

Historical Background

• There are a few historical mistakes that need to be fixed. The authors wrote on page 6: “It was within this crisis context that the Brazilian federal government, under the leadership of President Rodrigues Alves and the visionary sanitarian Oswaldo Cruz, initiated a monumental campaign to modernize public health. This effort culminated in the 1900 creation of the Instituto Soroterápico Federal in the district of Manguinhos, Rio de Janeiro, which would later be renamed Instituto Oswaldo Cruz (IOC) and become the cornerstone of the Fundação Oswaldo Cruz (Fiocruz) (8,9).”

Comment: The reader can believe that Rodrigues Alves and Oswaldo Cruz were both behind the creation of the Instituto Soroterápico Federal, which is wrong. The President of Brazil in 1899-1900 was Campos Salle, and the first director of the Instituto Soroterápico Federal was the Barão de Pedro Affonso. Rodrigues Alves was elected in 1902 under the promise of “sanitising” Rio de Janeiro, to which he placed Oswaldo Cruz at the head of the Diretoria Geral de Saúde Pública. But this came after the creation of the Instituto Soroterápico Federal, and not before as the authors seem to suggest.

Also, I would suggest removing the word “monumental” from the paragraph above. The modernisation of public health pushed by Rodrigues Alves and Oswaldo Cruz was important, but very limited to the city of Rio de Janeiro. Therefore, I suggest reworking this paragraph.

Again, on page 6: “From its inception, the institute was thrust onto the front lines of the plague fight. Charged with producing immunobiologicals, the institute’s first and most urgent mission was to manufacture a vaccine and serum against *Y. pestis* (4,8,9). Overcoming immense technical difficulties and initial scepticism, Oswaldo Cruz and his team, which included key figures like Vital Brazil, Carlos Chagas, Emílio Ribas, and Adolfo Lutz (10), successfully established large-scale production using equine hyperimmunization.”

Comment: The names cited in this phrase were not exactly part of Oswaldo Cruz’s team in charge of producing the anti-plague serum or vaccine, and the real collaborators of Oswaldo Cruz – Henrique Figueiredo de Vasconcellos and Cardoso Fontes – were not mentioned in this sentence. Carlos Chagas is correct, as he joined Manguinhos in the 1900s, but Vital Brazil, Emilio Ribas and Adolpho Lutz were part of São Paulo Public Health bureaucracy, and although they were friends with Oswaldo Cruz, and even some of them latter joined Manguinhos, I do not think this would be fair to simply put them as part of the “team” of Oswaldo Cruz, especially in this first moment of serum production.

Therefore, I suggest reworking this paragraph.

Classical Bacteriology Era

• In general, I found this era less developed than the others. Therefore, I would suggest expanding it a bit further by discussing two points: 1) MIOC was also a platform to discuss the tensions and unknowns of the anti-plague serum therapy, as evidenced by Arthur Moses’s 1914 article. 2) In 1922, Figueiredo de Vasconcellos wrote one of the first histories of plague in Brazil at the MIOC, arguing that studies on plague started a new scientific moment in Brazil.

Both articles should be mentioned.

Antibiotic Era/Antimicrobial Era.

• To start with, the authors use two terms to name this era: antimicrobial era in the text and antibiotic era on the timeline. I suggest choosing one and keeping it.

• But I do not think that either is the best term to describe this era of plague studies. Sure, antibiotics were a game-changer, but the articles at the MIOC were more interested in discussing how to fight plague, which involved several actions, from anti-rat campaigns to DDT pulverisations. Moreover, these articles were concerned on how to fight plague in rural areas. Therefore, I suggest something like the Rural Plague Era.

• Maybe the authors should mention, in a footnote, that the MIOC was not the main platform to publish about plague from the late 1920s until the 1970s. In this long period, two other journals were preferred by Brazilian and foreign scholars writing about plague in Brazil: the *Archivos de Hygiene* (in its different iterations) and the *Revista Brasileira de Malariologia*. To say that does not compromise the authors’ argument, but it will help the readers to look for other journals if they want to expand their comprehension of plague history in Brazil.

Final Remarks

These small mistakes and omissions are minor and can be easily fixed. Once they are done, I recommend the publication of the article.

## REVIEWER #2

The article presents all the aspects to publish in Memorias do Instituto Oswaldo Cruz.

For the referecences we suggest increase some papers published in “*História, Ciências, Saúde-Manguinhos*” (https://www.scielo.br/j/hcsm/grid ).

We strongly recommend some papers about scientifical missions in colonial and republican Brazil and the articles of Nisia Trindade Lima.

The following links can be used for research:

https://search.scielo.org/?q=*&lang=pt&count=15&from=0&output=site&sort=&format=summary&fb=&page=1&filter%5Bjournal_title%5D%5B%5D=Hist%C3%B3ria%2C+Ci%C3%AAncias%2C+Sa%C3%BAde-Manguinhos&q=*Miss%C3%B5es&lang=pt&page=1

https://www.scielo.br/j/hcsm/a/ssftpHJTrFMGJRkvg83nrYm/?lang=pt

## AUTHORS’ RESPONSE TO THE REVIEWERS

Response to the reviewers

Reviewer: 1

It gave me great pleasure to read and review the article “Memories of Memórias: shaping a century of plague research and public health policy in Brazil”. The abstract is well written and matches the text; the article is original and relevant to the field of the history of plague in Brazil; its methodology is coherent; the references are adequate, although they could be expanded; and, regarding the figures, the timeline contains a small typo, as described below.

In this article, the authors review and discuss a set of 17 articles on plague published at the MIOC, from its first edition in 1909 until the present day. One of the most interesting insights of the authors is their periodisation, dividing their sources into five main eras: “Classical Bacteriology Era, Antibiotic Era, Serological Era, Molecular Era, and Contemporary Era.” Drawing upon this periodisation, the authors then characterise each era, highlighting their main aspects and landmarks, the scientific shifts marking the end of the eras, and how the articles at the MIOC not only exemplify the eras’ trends, but most importantly, how they structured each of these eras. The authors are particularly suited to advance such a periodisation and analysis, as they wrote most of the articles of the last three periods of plague studies in Brazil. Therefore, I do not have any important comments on the Serological, Molecular, and Contemporary eras. Conversely, I have a few comments regarding the historical background, the Classical Bacteriology Era, and the Antibiotic Era, as outlined below.

Author: Please accept our sincerest gratitude for your thorough review and comments on our manuscript. We greatly appreciate the time and expertise you dedicated to our work. We have carefully considered all your suggestions and have incorporated them into the revised text. Below, you will find our detailed point-by-point responses, outlining how we have addressed each of your specific comments. Once again, thank you for your essential contribution to improving our work.

Reviewer: 1

Historical Background

1. There are a few historical mistakes that need to be fixed. The authors wrote on page 6: “It was within this crisis context that the Brazilian federal government, under the leadership of President Rodrigues Alves and the visionary sanitarian Oswaldo Cruz, initiated a monumental campaign to modernize public health. This effort culminated in the 1900 creation of the Instituto Soroterápico Federal in the district of Manguinhos, Rio de Janeiro, which would later be renamed Instituto Oswaldo Cruz (IOC) and become the cornerstone of the Fundação Oswaldo Cruz (Fiocruz) (8,9).”

The reader can believe that Rodrigues Alves and Oswaldo Cruz were both behind the creation of the Instituto Soroterápico Federal, which is wrong. The President of Brazil in 1899-1900 was Campos Salle, and the first director of the Instituto Soroterápico Federal was the Barão de Pedro Affonso. Rodrigues Alves was elected in 1902 under the promise of “sanitising” Rio de Janeiro, to which he placed Oswaldo Cruz at the head of the Diretoria Geral de Saúde Pública. But this came after the creation of the Instituto Soroterápico Federal, and not before as the authors seem to suggest.

Also, I would suggest removing the word “monumental” from the paragraph above. The modernisation of public health pushed by Rodrigues Alves and Oswaldo Cruz was important, but very limited to the city of Rio de Janeiro. Therefore, I suggest reworking this paragraph.

Author: We sincerely thank the reviewer for this correction. The chronological point raised is crucial for historical accuracy, and we have thoroughly revised the paragraph to ensure the sequence of events is correctly presented. As suggested, we have also removed the word “monumental”. The name of the Barão de Pedro Affonso has been appropriately included.

The corrected text now reads: “It was within this crisis context that the Brazilian federal government initiated a campaign to modernize public health (8). This effort culminated in the 1900 creation of the Instituto Soroterápico Federal (9) in the district of Manguinhos, Rio de Janeiro, under its first director, the Barão de Pedro Affonso. The institute would later be renamed Instituto Oswaldo Cruz (IOC), in recognition of Oswaldo Cruz’s role in Brazilian public health as the head of the Diretoria Geral de Saúde Pública, and become the cornerstone of the Fundação Oswaldo Cruz (Fiocruz) (10,11)”.

2. Again, on page 6: “From its inception, the institute was thrust onto the front lines of the plague fight. Charged with producing immunobiologicals, the institute’s first and most urgent mission was to manufacture a vaccine and serum against *Y. pestis* (4,8,9). Overcoming immense technical difficulties and initial scepticism, Oswaldo Cruz and his team, which included key figures like Vital Brazil, Carlos Chagas, Emílio Ribas, and Adolfo Lutz (10), successfully established large-scale production using equine hyperimmunization.”

The names cited in this phrase were not exactly part of Oswaldo Cruz’s team in charge of producing the anti-plague serum or vaccine, and the real collaborators of Oswaldo Cruz – Henrique Figueiredo de Vasconcellos and Cardoso Fontes – were not mentioned in this sentence. Carlos Chagas is correct, as he joined Manguinhos in the 1900s, but Vital Brazil, Emilio Ribas and Adolpho Lutz were part of São Paulo Public Health bureaucracy, and although they were friends with Oswaldo Cruz, and even some of them latter joined Manguinhos, I do not think this would be fair to simply put them as part of the “team” of Oswaldo Cruz, especially in this first moment of serum production.

Therefore, I suggest reworking this paragraph.

Author: We thank the reviewer for this correction. We have reworked the paragraph as suggested. The sentence now correctly credits Oswaldo Cruz’s immediate collaborators. The revised text reads: “Overcoming immense technical difficulties and initial scepticism, Oswaldo Cruz and his team, which included key figures like Henrique Figueiredo de Vasconcellos, Antônio Cardoso Fontes, and Carlos Chagas (12–14), successfully established large-scale production using equine hyperimmunization”.

3. Classical Bacteriology Era. In general, I found this era less developed than the others. Therefore, I would suggest expanding it a bit further by discussing two points: 1) MIOC was also a platform to discuss the tensions and unknowns of the anti-plague serum therapy, as evidenced by Arthur Moses’s 1914 article. 2) In 1922, Figueiredo de Vasconcellos wrote one of the first histories of plague in Brazil at the MIOC, arguing that studies on plague started a new scientific moment in Brazil.

Both articles should be mentioned.

Author: We are grateful for this excellent suggestion. In direct response to point 1, we have expanded the discussion to show how the MIOC served as a platform for scientific debate. We incorporated an analysis of Arthur Moses’s 1914 article and the uncertainties surrounding serum therapy at the time (please, see highlighted text in the main document).

In direct response to point 2, we have significantly developed the narrative to include Figueiredo de Vasconcellos’s 1922 historical account. We took the opportunity presented by this article to integrate a reference suggested by Reviewer 2, as its historical argument directly resonated with the broader historiography on scientific expeditions and nation-building. Specifically, we now show how Vasconcellos’s article served to canonize the institute’s early trajectory, thereby providing a primary source that anchors the later historical analysis of this formative period as a pivotal chapter for the construction of modern Brazilian science.

4. Antibiotic Era/Antimicrobial Era. To start with, the authors use two terms to name this era: antimicrobial era in the text and antibiotic era on the timeline. I suggest choosing one and keeping it. But I do not think that either is the best term to describe this era of plague studies. Sure, antibiotics were a game-changer, but the articles at the MIOC were more interested in discussing how to fight plague, which involved several actions, from anti-rat campaigns to DDT pulverisations. Moreover, these articles were concerned on how to fight plague in rural areas. Therefore, I suggest something like the Rural Plague Era.

Maybe the authors should mention, in a footnote, that the MIOC was not the main platform to publish about plague from the late 1920s until the 1970s. In this long period, two other journals were preferred by Brazilian and foreign scholars writing about plague in Brazil: the *Archivos de Hygiene* (in its different iterations) and the *Revista Brasileira de Malariologia*. To say that does not compromise the authors’ argument, but it will help the readers to look for other journals if they want to expand their comprehension of plague history in Brazil.

Author: We sincerely thank the reviewer for this insightful comment. We fully agree with the observation that the term antimicrobial/antibiotic Era does not fully capture the broader scope of the public health actions documented in the MIOC articles from this period. In direct response to this suggestion, we have adopted the proposed designation “Rural Plague Era” throughout the manuscript and in the timeline.

Furthermore, we have incorporated the contextual note regarding other key publication platforms. We have included this information directly in the text to conclude the section’s contextualization, rather than in a footnote. The added sentence reads: “It is relevant to note that from the late 1920s through the 1970s, much of the broader Brazilian plague research was also published in parallel journals, particularly the *Archivos de Hygiene* and the *Revista Brasileira de Malariologia*, reflecting the expansive and collaborative effort required to combat the disease during this period.”

5. Final remarks. These small mistakes and omissions are minor and can be easily fixed. Once they are done, I recommend the publication of the article.

Author: We sincerely thank the reviewer for the positive final assessment and for the recommendation for publication.

Reviewer: 2

1. The article presents all the aspects to publish in Memorias do Instituto Oswaldo Cruz.

Author: We thank Reviewer 2 for the positive assessment. We have carefully considered the specific suggestions provided and have incorporated them into the revised manuscript.

2. For the referecences we suggest increase some papers published in “História, ciências, Saúde” (https://www.scielo.br/j/hcsm/grid ).

We strongly recommend some papers about scientifical missions in colonial and republican Brazil and the articles of Nisia Trindade Lima.

The following links can be used for research: https://search.scielo.org/?q=*&lang=pt&count=15&from=0&output=site&sort=&format=summary&fb=&page=1&filter%5Bjournal_title%5D%5B%5D=Hist%C3%B3ria%2C+Ci%C3%AAncias%2C+Sa%C3%BAde-Manguinhos&q=*Miss%C3%B5es&lang=pt&page=1 and https://www.scielo.br/j/hcsm/a/ssftpHJTrFMGJRkvg83nrYm/?lang=pt.

Author: We thank the reviewer for the suggestion. We agree that these studies provide essential historical and historiographical context for understanding scientific missions and the construction of public health in Brazil. In direct response to the recommendation, we have incorporated three key references from this line of research into our bibliography:

1. We included the article “Não é meu intuito estabelecer polêmica”: a chegada da peste ao Brasil, análise de uma controvérsia, 1899 (Nascimento and Silva, 2013), published in *História, Ciências, Saúde – Manguinhos*, that offers a detailed analysis of the public debate that preceded the creation of the Instituto Soroterápico Federal, contextualizing the political pressure for a scientific response.

2. We also added the study “A peste bubônica no Rio de Janeiro e as estratégias públicas no seu combate (1900-1906)”, published in the journal *Territórios e Fronteiras*, by Dilene Raimundo do Nascimento, of the Casa de Oswaldo Cruz/Fiocruz, to better detail the public health crisis scenario following the arrival of the disease.

3. Specifically following the indication regarding the work of Nísia Trindade, we included her article “Missões civilizatórias da República e interpretação do Brasil”, published in *História, Ciências, Saúde – Manguinhos*. We took the opportunity presented by the inclusion of Figueiredo de Vasconcellos’s 1922 historical article (Reviewer 1) to draw a parallel between the internal narrative of the MIOC and broader historiographical interpretations of scientific expeditions as nation-building projects, citing this work.

We believe that incorporating these references in a contextual and integrated manner into our narrative addresses the spirit of the suggestion and adds a valuable layer of historical depth to our study, without diverting the primary focus of the analysis on the *Memórias do Instituto Oswaldo Cruz* periodical. We thank the reviewer again for the pertinent bibliographic guidance.

---

## [Reviewer Report · REVIEWERS COMMENTS]

## REVIEWER #1

The authors have fully answered my comments, and I am pleased to recommend the acceptance of the article. Just a minor comment: notes 13 and 14 are the same, and one needs to be excluded. Otherwise, it looks fine.

## REVIEWER #2

No comments.